# Reimagining the Role of Health Departments and Their Partners in Addressing Climate Change: Revising the Building Resilience against Climate Effects (BRACE) Framework

**DOI:** 10.3390/ijerph20156447

**Published:** 2023-07-26

**Authors:** Stephenie C. Lemon, Heather A. Joseph, Samantha Williams, Claudia Brown, Semra Aytur, Katherine Catalano, Stacey Chacker, Karin V. Goins, Linda Rudolph, Sandra Whitehead, Sara Zimmerman, Paul J. Schramm

**Affiliations:** 1Prevention Research Center, UMass Chan Medical School, Worcester, MA 01655, USA; karin.goins@umassmed.edu; 2Climate and Health Program, Centers for Disease Control and Prevention, Atlanta, GA 30333, USA; hbj7@cdc.gov (H.A.J.); pog3@cdc.gov (S.W.); wyn3@cdc.gov (C.B.); imw3@cdc.gov (P.J.S.); 3Department of Health Management and Policy, College of Health and Human Services, University of New Hampshire, Durham, NH 03824, USA; semra.aytur@unh.edu; 4Center for Climate, Health and Equity, American Public Health Association, Washington, DC 20001, USA; katherine.catalano@apha.org; 5Health Resources in Action, Boston, MA 02116, USA; schacker@hria.org; 6Center for Climate Change and Health, Public Health Institute, Oakland, CA 94607, USA; rudolph.linda@gmail.com; 7College of Professional Studies, Sustainable Urban Planning Program, The George Washington University, Washington, DC 20052, USA; swhitehead@email.gwu.edu; 8Climate Equity Policy Center, Berkeley, CA 94702, USA; sara@climateequitycenter.org

**Keywords:** climate, health departments, mitigation, adaptation, community engagement, equity

## Abstract

Public health departments have important roles to play in addressing the local health impacts of climate change, yet are often not well prepared to do so. The Climate and Health Program (CHP) at the Centers for Disease Control and Prevention (CDC) created the Building Resilience Against Climate Effects (BRACE) framework in 2012 as a five-step planning framework to support public health departments and their partners to respond to the health impacts of climate change. CHP has initiated a process to revise the framework to address learnings from a decade of experience with BRACE and advances in the science and practice of addressing climate and health. The aim of this manuscript is to describe the methodology for revising the BRACE framework and the expected outputs of this process. Development of the revised framework and associated guidance and tools will be guided by a multi-sector expert panel, and finalization will be informed by usability testing. Planned revisions to BRACE will (1) be consistent with the vision of Public Health 3.0 and position health departments as “chief health strategists” in their communities, who are responsible for facilitating the establishment and maintenance of cross-sector collaborations with community organizations, other partners, and other government agencies to address local climate impacts and prevent further harm to historically underserved communities; (2) place health equity as a central, guiding tenet; (3) incorporate greenhouse gas mitigation strategies, in addition to its previous focus on climate adaptation; and (4) feature a new set of tools to support BRACE implementation among a diverse set of users. The revised BRACE framework and the associated tools will support public health departments and their partners as they strive to prevent and reduce the negative health impacts of climate change for everyone, while focusing on improving health equity.

## 1. Introduction

Climate change is a ‘crisis multiplier’ [1], as its harmful environmental effects cause direct and indirect impacts on human health, which intersects with other preexisting drivers of health inequities, such as poverty and structural racism. Health impacts include, but are not limited to, food-, water-, and vector-borne diseases; pulmonary and cardiovascular diseases; undernutrition; adverse reproductive outcomes; mental health disorders; heat-related illness; and extreme weather-associated injury and death [2,3,4,5,6,7,8,9]. Recent events, including the COVID-19 pandemic [10], wildfires and poor air quality [11,12], droughts [13], and extreme weather events [14,15], underscore the importance of recognizing the synergistic effects of climate change. The economic costs of climate change are staggering. It has been estimated that the domestic health costs of climate change and fossil fuel-related air pollution exceed USD 800 billion each year, stemming from the impacts of premature deaths, hospitalizations, serious injuries, mental health ailments, lost wages, and missed days of work, among other problems. These costs will increase as the climate continues to warm [16]. 

Like many societal problems, the effects of climate change are not distributed equally [17,18]. Impacts vary by exposure, sensitivity, and adaptive capacity. Those at higher risk for adverse effects include persons with low socioeconomic status, certain racial and ethnic groups, older adults, children, persons with disabilities or preexisting health conditions, and workers exposed to environmental hazards, such as heat, air pollution, or wildfire smoke. Additionally, people living in different geographic regions [19], including urban vs. non-urban settings [20], face varying combinations of climate threats, such as wildfire, drought, extreme temperatures, sea level rise, and flooding. The inequitable distribution of climate change vulnerability is a product of the same factors that lead to other poor health outcomes, such as poverty, poor housing, unsafe or unhealthy environmental conditions, and lack of access to good jobs, quality education, and comprehensive health care, which are fundamentally driven by racism, discrimination, stigma, and disenfranchisement [21]. Considering that almost half of U.S. adults already report medical financial hardship, climate change will continue to act as a crisis multiplier by worsening both health and economic burdens, with a high potential to exacerbate inequities for the most vulnerable [16]. 

### 1.1. The Role of Public Health in Addressing Climate Change

Calls for public health engagement in climate change solutions have accelerated [22,23]. Climate change has been prioritized by the nation’s foremost public health organizations as a leading health crisis [24,25,26,27]. As well-positioned messengers of health issues in their jurisdictions, state, tribal, local, and territorial (STLT) public health officials can lead collaborative efforts addressing the causes and local impacts of climate change. In fact, these officials may be more effective than national authorities at conveying information about climate change since its effects are experienced locally [22,28].

However, public health departments are generally not well prepared to prevent and address climate threats [29,30,31]. Most public health departments, particularly at the local level, face financial constraints [32]. Public health departments are often not optimally aligned to participate in larger community climate change planning, which may be led by a sustainability, environmental services, and/or planning department, or an executive office [33]. Existing climate efforts within health departments are often siloed [34], underfunded [35,36], and under-resourced [37], and may lack capacity to adequately address inequities [38]. Public health departments frequently rely on grant funding that is designated for singular programmatic responses to specific infectious or chronic health conditions, with limited latitude to address root causes or synergistic drivers of ill health [37]. Workforce capacity issues [39] and a general lack of environmental health literacy among the U.S. population further compounds these challenges [40]. To accelerate climate action in their jurisdictions, many health departments need support to increase their internal capacity, as well as collaborations across units, government departments and offices, and community organizations, especially those that represent people disproportionately affected by climate change [41,42,43,44]. 

### 1.2. The Building Resilience against Climate Effects (BRACE) Framework

To assist STLT departments to develop strategies and programs to help communities prepare for and respond to the health effects of climate change, the CDC’s Climate and Health Program (CHP) developed the Building Resilience Against Climate Effects (BRACE) framework [45]. The initial intent for the framework was to enable STLT public health practitioners to use the best available science to project likely health impacts due to climate change within their jurisdiction and prioritize, implement, and evaluate interventions. 

The BRACE framework is an iterative process comprising five sequential steps (see Figure 1):Step 1: Anticipate Climate Impacts and Assess Vulnerabilities

Identify the scope of climate impacts, associated potential health outcomes, and populations and locations vulnerable to these health impacts.

Step 2: Project the Disease Burden

Estimate or quantify the additional burden of disease outcomes associated with climate change.

Step 3: Assess Public Health Interventions

Identify the most suitable health interventions for the identified health impacts of greatest concern.

Step 4: Develop and Implement a Climate and Health Adaptation Plan

Develop a written adaptation plan that is regularly updated. Disseminate and oversee implementation of the plan.

Step 5: Evaluate Impact and Improve Quality of Activities

Evaluate the process to determine the effect of the adaptation action and ways to improve it. 

The BRACE framework has been implemented by grant recipients of the CDC’s Climate-Ready States and Cities Initiative since 2012 [28,46]. This has funded 20 health departments to take action to prepare for climate change across the nation. 

In response to needs for further support to implement the BRACE framework identified by CDC grant recipients of the Climate-Ready States and Cities Initiative, the CDC developed and disseminated the Climate and Health Technical Report Series [47]. This included reports on topics related to the individual steps of the BRACE framework, such as climate modeling and projections, assessing health vulnerability to climate change, projecting disease burden, and identifying and evaluating climate and health interventions. In addition, the CDC developed and disseminated case studies, webinars, videos, templates, and communication materials in support of the Technical Report Series and other priority topics that arose. 

As justice, equity, diversity, and inclusion principles are crucial to climate change adaptation planning, the American Public Health Association (APHA) also created, with CDC funding and collaboration, *Climate Change and Health Playbook: Adaptation Planning for Justice, Equity and Inclusion* [48]. This resource was designed to supplement the BRACE framework and provides guidance on integrating justice, equity, diversity, and inclusion into climate and health efforts. 

A modified version of the BRACE framework (I-BRACE), which was developed by one tribe to incorporate indigenous concepts of health and values related to data and decision-making, has been developed and implemented by other tribes through the CDC’s Climate-Ready Tribes Initiative since 2016 [49], which has supported ten federally recognized tribes. 

### 1.3. Identified Gaps with BRACE Framework 

Since the launch of the BRACE framework, public health practitioners have come to understand the strengths and limitations of the model. CHP has consistently engaged grant recipients in facilitated discussions during site visits, grantee meetings, and technical assistance calls about what is working well and what improvements could enhance implementation [50]. 

CHP’s grant recipients have shared that BRACE led to increased public engagement, helped mainstream the topic of climate and health, and offered the structure and direction needed to accomplish their objectives [28]. However, there were also limitations and impracticalities in implementing the BRACE framework. Recipients voiced that the early steps of BRACE required technical capacity beyond what a typical health department could do, even with dedicated climate and health funding. Recipients also described staffing challenges [28] related to managing all the different components of BRACE, which could entail an epidemiologist, statistician, geographic information system specialist, program implementation specialist, community engagement specialist, and evaluator. Further, recipients highlighted that BRACE was not tailored to their organization’s resources and capacity, nor the level of government (i.e., state vs. local) in which they operate, limiting their effectiveness in implementing BRACE. Recipients also highlighted the need for additional practical guidance for each step, as well as other key components of climate and health adaptation, such as partnership development, authentic community engagement with appropriate funding support, communications, and navigating political contexts [51]. It is also recognized that the experiences of states, cities, and tribal partners may differ and that guidance specific to each is needed [28,37,51]. 

### 1.4. Aim

In response to these identified gaps, along with the emerging evidence base documenting the importance of the climate in health and associated inequities, CHP initiated a comprehensive revision to the BRACE framework. The aim of this manuscript is to describe the methodology for revising the BRACE framework and expected outputs of this process. 

## 2. Methods

### 2.1. Planned Revisions to the BRACE Framework

CHP envisioned several enhancements, which included alignment with the concept of Public Health 3.0 [52], emphasis on health equity, inclusion of mitigation strategies, and greater application to diverse settings. These priorities were selected based on feedback from grant recipients and implementation partners, as well as gaps identified by CHP staff through years of practice. A more detailed explanation of priority enhancement areas are as follows. 

#### 2.1.1. Alignment with Public Health 3.0 

The revised BRACE will be consistent with the vision of Public Health 3.0, which is the “next generation” approach for health departments and emphasizes “partnering across multiple sectors and leveraging data and resources to address social, environmental, and economic conditions that affect health and health equity [52].” This approach places public health departments as “chief health strategists” in their communities, responsible for leading efforts, yet recognizing the strengths and value of cross-sector collaborators in achieving common goals. The next generation of BRACE will charge public health departments with better understanding their local landscape regarding climate change efforts in order to align and/or integrate their use of the revised framework with existing efforts. This can be done in collaboration with community organizations focused on addressing climate and environmental justice or health equity more broadly; with other state and local departments, such as planning, sustainability, and public works; and with a range of other partners in a manner that best works for their local context. 

#### 2.1.2. Emphasis on Health Equity 

The revised BRACE will place health equity as a central, guiding tenet. Public health departments and their partners will be charged with identifying and collaborating with communities who are disproportionately impacted by local climate effects as part of BRACE implementation. As described above, BRACE now includes supplemental guidance on integrating equity via the *Climate Change and Health Playbook: Adaptation Planning for Justice, Equity, Diversity and Inclusion* [48], which was created by the APHA with funding and technical support from CHP. However, the *Playbook* was developed as a companion to BRACE, rather than a core component of BRACE implementation. The revised BRACE will instead directly incorporate equity principles and provide practical guidance on prioritizing equity in each of the components that comprise the revised framework. 

#### 2.1.3. Inclusion of Climate Change Mitigation Strategies 

Whereas the previous version of BRACE focused only on supporting public health departments to address adaptation to climate change, i.e., reducing vulnerability to the adverse health effects of climate change, the revised BRACE will also include guidance on mitigation, i.e., reducing greenhouse gas emissions to slow the pace of climate change. This will include a special focus on locally relevant mitigation strategies that have the capacity to produce direct health co-benefits for community members (e.g., increased physical activity associated with active transportation) [53]. Consistent with Public Health 3.0 [52], making BRACE inclusive of mitigation will require cross-sector collaboration with local community and grassroots organizations focused on the priority issue and across other government agencies (e.g., transportation, planning, and public works) and with other divisions within the health department working on these issues (e.g., community health staff). 

#### 2.1.4. Application to Diverse Settings

In addition to a revision of the framework, the revised BRACE will include a new set of tools that will support the implementation of BRACE and meet the needs of a diverse set of users. Limited implementation guidance is a noted challenge of using BRACE [51]. Tools that support implementation of the revised BRACE will require careful attention to the complexities of implementing each step in a localized, multi-sectoral, collaborative context and provide practical guidance to implementers. To be useful, implementation guidance should focus on guiding communities to understand their local context and to implement the revised BRACE framework and associated interventions to fit this context. Factors such as climate risks, political will, user type (e.g., state, territorial, city, tribal), resource availability, community priorities, and public support related to climate action should be considered. Tools will also be provided to support public health departments with varying capacity and resources. For example, tools will provide guidance on finding resources that can be leveraged to forecast climate impacts and project disease burden, rather than expecting health departments to collect their own data. Guidance to support BRACE implementation will thus be comprehensive and practical, yet flexible enough to be tailored to local circumstances.

To lead the revision to the BRACE framework, CHP awarded a cooperative agreement to researchers at the Prevention Research Center at UMass Chan Medical School, which assembled a multi-disciplinary team representing climate-and-equity focused agencies (Climate Equity Policy Center), public health institutes (Health Resources in Action, the Public Health Institute), a professional society (American Public Health Association), and academic institutions (University of New Hampshire, George Washington University) to lead this effort. The objective is for this next version of BRACE to support users focused on building community resilience, promoting public health, and reducing health inequalities through climate change mitigation and/or adaptation activities. The primary intended users of the revised BRACE are STLT departments in partnership with community organizations, other government agencies, and any other entity with a vested interest in addressing climate change, health, and equity in their communities. The revision will be completed over a two-year period (September 2022–September 2024).

### 2.2. Methodology to Revise the BRACE Framework and Create Implementation Tools

The development of the revised BRACE framework will be informed by a series of rapid research methods, shown in Figure 2 and described in detail in Section 2.2.1, Section 2.2.2, Section 2.2.3 and Section 2.2.4.

Rapid research methods aim to address the need for cost-effective, efficient, and timely results in rapidly changing dynamic situations [54], enabling actionable feedback and flexibility. As the primary intended users of the revised BRACE framework are public health departments and their community and other partners, all activities to inform the BRACE revision will involve engaging experts who represent these groups.

#### 2.2.1. Expert Panel

Development of the revised framework and associated implementation tools will be informed by an Expert Panel. The 32-member Expert Panel was selected to represent a range of potential users of the revised framework and/or because of their expertise related to climate and health. Members include representatives of the following: nine state health departments, two county/city health departments, two tribal health departments, six community organizations, two federal agencies, eight national organizations, and three academic organizations. Of the 32 members, 10 are current or former recipients of the CDC’s Climate-Ready States and Cities Initiative and/or the Climate-Ready Tribes Initiative, 5 are unfunded users of the current BRACE framework, 3 had roles working with the CDC on BRACE as either a developer or evaluator, and 14 had no experience with BRACE. Members represent 19 U.S. states and the District of Columbia. Individuals invited to be on the panel were identified by members of the project team and CHP staff. 

#### 2.2.2. Approach to Creating the Revised BRACE Framework

We will first revise the BRACE framework using a three-phase Modified Delphi approach [55]. This approach is a well-established method for working with a team of experts to achieve consensus on a topic of interest. The Modified Delphi approach used to revise the BRACE framework will entail the Expert Panel participating in three phases: (1) formative, (2) development, and (3) refinement. 

The formative phase will consist of gathering information from the Expert Panel members to inform the framework revision. Key informant interviews will be conducted with each Expert Panel member. These interviews will be designed to understand panel members’ thoughts and experiences (when applicable) with the BRACE framework and solicit input on how to update the framework in a manner consistent with the priorities listed above. Questions will ask about participants’ experiences with BRACE (e.g., what worked well and what did not), as well as their perceptions of specific steps within the original BRACE framework. Questions about collaboration, barriers, opportunities, and new ideas about centering equity and including mitigation in the revised BRACE framework will also be included. Rapid qualitative analysis (RCA) techniques used in implementation science will be leveraged to produce timely, actionable results based on inputs from the experts in a manner that balances rigor and efficiency [56,57]. 

The development phase will consist of drafting the revised framework based on the results of the key informant interviews, with input from the project team members and staff at CHP. The refinement phase will consist of utilizing the community engagement studio methodology with the Expert Panel [58,59]. Studios are semi-structured, facilitated group discussions, in which the experts are first presented with a description of the draft revised framework and then asked a series of questions to capture their reactions. The facilitated discussion will be designed to obtain feedback on how to improve the wording and presentation, and to assess how the framework is received in terms of perceived acceptability (satisfaction, palatability), perceived appropriateness (fit, relevance, or compatibility), and perceived feasibility (likelihood of implementation in practice) [60]. The results will be used to further refine the framework. 

#### 2.2.3. Approach to Creating Implementation Tools

Once the framework has been revised, we will conduct a second three-phase Modified Delphi approach [55] with the Expert Panel to create tools to support its implementation, similarly consisting of (1) formative, (2) development, and (3) refinement phases. In the formative phase, we will collect information that informs the format and content of the implementation tools. The nominal group technique (NGT) will be used to solicit from the Expert Panel members their perceptions of the main capacity-building needs of public health departments and their partners regarding implementation of the revised BRACE framework. NGT is a structured group process that is designed to generate a prioritized list in response to a specific prompt or question [61]. We will query about the components included in the revised BRACE framework. For each component, participants will address a discussion prompt. For example, a prompt could include: “What types of knowledge, skills and/or supports do you think health departments and their partners will need to collaboratively implement component X of the BRACE framework?”. They will then engage in a round robin style approach to generating ideas, followed by facilitated discussion, and, finally, a ranking and rating process to prioritize what they believe are the most important knowledge, skills, and/or supports required. In the development phase, a semi-structured approach delineated by the implementation mapping process will be used [62]. This will consist of (1) creating performance objectives for each element of the revised BRACE framework, (2) mapping specific strategies/formats to the objectives, (3) developing the overall structure of the strategies/formats, and (4) creating detailed content. We envision tools with “basic” and “enhanced” options that can be used by lower- and higher-resourced public health departments and communities. We also envision tools that can support different collaborators based on their roles in implementing BRACE [63], with public health departments serving as the lead convener and coordinator and/or as a collaborator in their BRACE efforts; community organizations leading activities that play to their strengths, such as activities focused on equity promotion and community engagement; and CHP funding the Climate-Ready States and Cities Initiative and providing technical assistance activities. Development will also include creating a brand identity that will be used in all revised BRACE components. Once fully drafted, we will refine the implementation tools using community engagement studios, following an approach similar to the one described above. 

#### 2.2.4. Usability Testing

Usability-like testing of the combined revised framework and implementation tools will be conducted to better understand their utility for real-world practice [64]. The usability testing will be conducted with teams representing 2–3 jurisdictions that were not represented on the Expert Panel. Within each jurisdiction, 3–4 individuals, including 1 health department lead and 2–3 partners, will be recruited. Group usability testing sessions will be held separately for each jurisdiction. Through a facilitated process, participants will be presented with the revised BRACE components and associated implementation tools. They will be asked to reflect on a series of questions about how they could envision using the components of the revised framework and associated tools, what they like about them, and how they could be improved to better meet their needs. Prompts will ask participants to reflect on the acceptability, appropriateness, and feasibility [60] of each step and tool. Information gleaned from these sessions will inform final modifications to the revised framework and implementation tools.

## 3. Discussion

At the conclusion of this project, a revised BRACE framework and associated implementation tools that can be used by STLT health departments and their partners will be finalized. This revised BRACE package will be disseminated widely via the CDC website and promoted through presentations, academic publications, and other means. The revised BRACE framework and associated implementation tools will be incorporated into future CHP funding opportunities and be available to non-funded STLT public health departments, their community and government partners, and others to guide their climate and health work. 

The revised BRACE framework and the associated implementation tools are intended to guide the work of STLT public health departments and their partners. The revision will prioritize four enhancements to the current BRACE framework. First, consistent with the vision of Public Health 3.0 [52], community and cross-sector collaborations will be integral to effectively addressing climate and health. The revised BRACE framework will place partnerships and community engagement as foundational to climate-related work of STLT public health departments. Second, the revised framework also includes health equity as a foundational element and prioritizes partnering with communities and organizations that represent communities who are disproportionately affected by the health consequences of climate change. Third, the revised framework will support climate mitigation activities, along with climate adaptation activities. Lastly, the implementation tools for the revised BRACE framework will strive to meet the needs of a range of potential end-users [28,37,50]. Other potential revisions will be identified through the input of the Expert Panel, which was purposefully composed of individuals who represent varied experiences and constituencies (e.g., federal, state, local, tribal; urban, non-urban), resource/capacity levels, climate risks, and political wills. 

Users of the revised BRACE framework and associated implementation tools should also be encouraged to use these products in a flexible manner that allows them to respond to climate threats and incorporate promising practices that meet local and/or regional needs. Although many of the health effects of climate change are felt locally, some effects, such as smoke from wildfires, are affecting large geographies (even across national borders), requiring regional adaptation/mitigation approaches. STLT health departments and their partners across the nation are recognizing the importance of collaborating regionally on these issues [65,66,67]. Future research should explore the utility of the revised BRACE in supporting regional collaborations and should also continue to examine the full suite of barriers and enablers pertaining to health departments’ support and promotion of climate change adaptation and mitigation efforts.

There are limitations to the methodology used in this project that must be acknowledged. Resource limitations restricted the Expert Panel to 32 members. Although we attempted to include a diverse group of potential BRACE users, it is possible that important perspectives may be missing from this revision process. For example, we were also unable to include youth perspectives. In addition, this project is limited to garnering immediate feedback on the development and initial usability testing. Future research should continue to explore end-users’ perceptions of the utility of the revised BRACE framework and implementation tools over time [68]. 

## 4. Conclusions

STLT public health departments have an important role to play in addressing the health impacts of climate change in local communities, as both leaders and collaborators. The CDC’s CHP is committed to supporting STLT health departments’ engagement in collaborative work to address the health impacts in their communities. The revised BRACE framework and the associated implementation tools will provide the foundation for the next generation of this nation’s public health-focused climate mitigation and adaptation efforts, informing the work of communities across the country as they strive to prevent and minimize the negative health impacts of climate change. 

## Figures and Tables

**Figure 1 ijerph-20-06447-f001:**
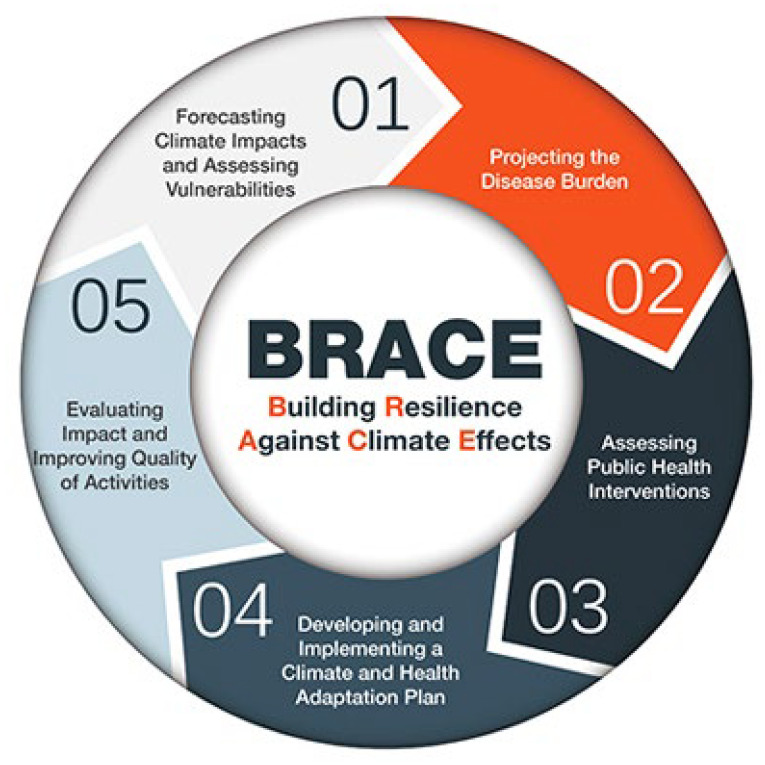
The five steps of the existing BRACE Framework.

**Figure 2 ijerph-20-06447-f002:**
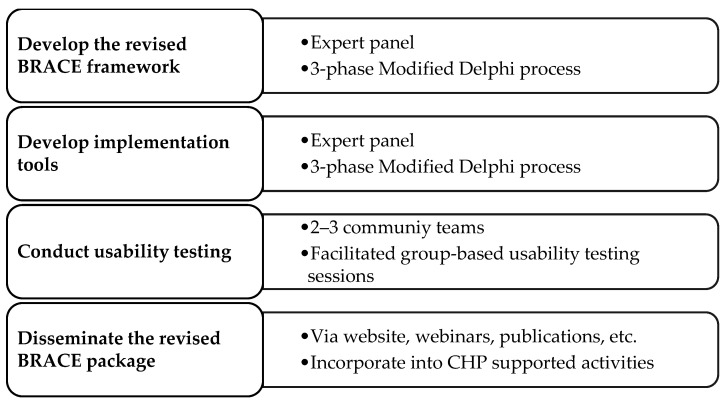
Methodology to develop, pilot test, and disseminate the revised BRACE framework and implementation guidance and tools.

## Data Availability

No new data were created or analyzed in this study. Data sharing is not applicable to this article.

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
