# Peer review of "Reimagining the Role of Health Departments and Their Partners in Addressing Climate Change: Revising the Building Resilience against Climate Effects (BRACE) Framework"

_ijerph, 2023, doi:10.3390/ijerph20156447_

Round 1
Reviewer 1 Report
This study protocol manuscript aims to outline the context, approach and qualitative analytical methods for revising and updating the US CDC’s Climate and Health Program’s (CHP) successful Building Resilience Against Climate Effects (BRACE) framework and its implementation tools, based on ten years of implementation experience with grantees, reviews of the program, and other inputs such as a landscape review of the climate adaptation framework literature.
This manuscript is therefore a highly welcome and extremely important addition to the climate change and health field, and to the future of local public health approach to climate change in the US in the coming years. In its current form it presents the core guiding concepts and methods envisioned for this revision of the BRACE framework, which seem generally sound and appropriate; the strong focus on equity is particularly noteworthy. However, given its importance to the field, the paper could benefit from being developed in a slightly clearer and more authoritative way. My main comments to this effect are noted below, followed by line-by-line comments by section.
Main comments:
1. A clear one-sentence goal statement for this paper is currently lacking and is important to include, ideally in the introduction before the Methods section. It should also be present in the Abstract.
2. Several long sections of the paper are without references and it is difficult to know on what basis the statements are being made, making these sections seem somewhat anecdotal rather than authoritative. Particularly lines 138-168 (learning and evaluation of BRACE), lines 170-212 (landscape review of frameworks), and lines 229-281 (“enhancements”). The impression is of a viewpoint paper rather than a scholarly, data-driven evaluation and roadmap for review and revision. Detailed comments below provide some suggestions for remedying this. Two points are particularly notable:
(i) The landscape review of the climate adaptation framework literature reads almost like a separate independent paper, suggest authors consider this as an option. If so, that separate landscape review could be summarized briefly and reference in this paper.
(ii) The genesis of the four “enhancements” to BRACE is not provided. These four enhancements, which appear highly valid and worthy, form the grounding substantive guidance for the new BRACE framework. Yet the way in which these four were selected, and on what basis, is not evident from the paper. Why these four and not others?
3. This paper does not seem to consider several climate and health challenges that have recently grown substantially in importance for local US health departments. These include COVID-19 (lessons for local health departments engagement on climate, interactions with air quality, etc.), wildfire smoke exposure (in the US West and now East as well with Canadian wildfires), the major impacts of drought, particularly in the US West. Etc.
4. There is little attention to the distinction between urban and non-urban areas generally in the paper. Cities and states as well as tribes have been grantees under BRACE, and their capacities, resources, and experiences have differed. Given the large urban population affected by climate change (and the focus of many of the reviewed frameworks on cities), brief words on the urban experience, and specific framework-related issues for cities under the new BRACE would be helpful.
5. The Methods section could be more clearly presented and organized. For example, Figure 2 belongs to this section, however, appears in the Introduction, and is currently not referenced or discussed in the paper, while this Figure and the Methods text are not mutually consistent.
6. What are the expected outputs and timeframe of this study protocol work? Will the iterative adaptive stepwise framework remain as an underpinning? Will the results be published, and in what way?
Specific comments:
Title
“Reimagining the role of PH departments… etc.” suggests the aspirational goal for PH departments to take on greater leadership for climate change policy and practice as a cornerstone of the new BRACE program. In this reviewer’s opinion, this vision is highly welcome and justified. However, the paper’s content does not sufficiently identify the reasons why PH departments are often sidelined from climate policy and practice currently; nor do the new pathways proposed for BRACE fully explain why public health departments might be in a different, better position with the new BRACE. Therefore, the paper does not seem to deliver yet on this title. Suggest a somewhat fuller diagnosis of why these departments currently are excluded from climate change efforts and how the proposed revised approach may change this, based on the large available literature (or modifying the title for a better fit with paper content).
Abstract
What is the goal of this paper? It is not stated. I understand it to be (in simple, brief form): outlining the protocol for the revision of the BRACE framework and its tools. Please define clearly the goal of this submitted manuscript.
Introduction
Line 46-47. Would be useful to refer to air quality from wildfires which have substantially heightened climate and health awareness, particularly following COVID-19, and to the WHO estimates of annual deaths from air pollution. The climate and health effects in this section generally could be somewhat updated to reflect more recent learning and current/emerging concerns, including for example, widespread exposure to wildfire smoke.
Line 71-72. Many climate health effects are indeed felt locally. However, there are also others (like the above wildfire smoke example, but also for example drought) which may be felt regionally, and may require regional approaches. Given the growing challenges of air quality and water quality/quantity, incorporating into the new BRACE ways for local health departments to collaborate regionally on such topics may be of growing importance.
Line 76-73. It would be helpful to provide a clearer explanation of the problem stated that PH departments are “often not optimally aligned… etc.” and are “siloed, underfunded… etc.” The evidence from the literature for these statements is vast, but the citations for this literature do not seem to currently be presented in a helpful or specific way (with most added to the end of the para). Since this paper is intended to set out the approach to the new framework for BRACE, it is should be authoritative and grounded in the literature.
Line 112-113. The difference between experience of states and cities in the BRACE program should be at least mentioned, and ideally somewhat expanded upon. This could carry through to the enhancements priorities and qualitative review for the new BRACE in subsequent sections of the paper -- i.e., will there be differences in approach and tools of the new BRACE framework for states vs cities (vs tribes). This is hinted at currently in the fourth "enhancement" priority area of "implementation in diverse settings," but not sufficiently developed as a theme.
Section 1.2. Identified gaps and other frameworks. This section would be more effective if it separated the two topics.
Lines 138-159. This section on “identified gaps” provides useful information on grantee experience. However, the whole section is unreferenced. Grant recipient feedback is generally mentioned as the source of these findings, including from phone calls and site visits. But no information is provided on the documented sources for these grantee feedback findings. Multiple documentation on BRACE grantee views have been made public over the years. Please add references to the documented sources for this feedback. Currently this section reads as too anecdotal for an authoritative piece, and does not provide enough clarity on where the “learning” is coming from.
It is this reviewer’s understanding that CHP/BRACE has also been formally reviewed, and these formal reviews should also be mentioned within the “identified gaps” section. E.g., there was at least one expert workshop sponsored by CDC, to this reviewer’s knowledge, and likely more. The review findings should be mentioned, incorporated and referenced, even if briefly.
Lines 160-212. This section on “other frameworks” should be treated separately from “identified gaps” for clarity. This whole section is also unreferenced, and reads as somewhat anecdotal. Was the “landscape review” of climate adaptation frameworks conducted separately and produced as a report? If so, that report should be cited. Or, is this paper serving to report the findings for that landscape review? If this is the case, in essence then this paper has two goals, one a climate health framework study/review protocol and the other a landscape review of climate and health frameworks. At the very least this additional goal should be clarified in the manuscript goal statement, and the key findings reported in a way that is more concise and directly relevant to the main paper goal, i.e., the protocol for development of the new BRACE.
However, another option the study team could consider would be to produce the climate and health frameworks landscape review as a separate review paper. Such a paper would allow expanded development of the findings of the landscape review which would be a good addition to the climate and health literature. That landscape review paper could then be briefly summarized and referenced in the present BRACE revision study protocol paper, with the outcome of helpfully sharpening the focus of the current BRACE revision protocol paper. The fact that the framework landscape review is an input to redesign of BRACE suggests it is well worth doing it as a comprehensive review and separate published manuscript.
Line 184-186. The landscape review search string involved the keyword “adaptation” therefore it is not surprising adaptation was a core concept in the frameworks identified, and mitigation was not. The wording in these lines thus should be adjusted to make this clear. (But -- why not have broadened the search words to identify other frameworks not focused on adaptation only e.g., “health” and “mitigation”? Other frameworks dealing with mitigation might have emerged…)
Importantly, and as noted for the Abstract, the paper currently lacks, and needs, a clear goal statement at the end of the introduction right before the Methods. “This paper aims to …”
Methods
The four “enhancements” to BRACE seem clear and relevant. But where did they come from? Why these and not others? A brief initial paragraph is needed to explain the genesis of these enhancement priorities, and the process of choosing these four from among many potential other enhancements. One gathers from the current text that this emerged from BRACE review recommendations, grantee feedback, review of the literature, review of other frameworks. But a short intro sentence/paragraph to describe the choice is important to make this clear.
As a general comment, as noted above, additional clarity on state, city or tribal/territorial grantees and the differences in needs, challenges and conditions in each could be more adequately addressed in this paper. For example, in the “enhancements” one might expect to see some guiding principles for grants to cities, given many of the frameworks in the landscape review address urban areas.
Line 282. The methodological approach to revising the BRACE framework seems sounds and justified, and the specific qualitative methods to be appropriate. But this section is presented in a way that is somewhat difficult to follow currently. Specifically, Figure 2 (labeled as “methodology to develop, pilot test and disseminate the revised BRACE framework…”) is unanchored to the text, and appears in the Introduction rather than Methods section. Figure 2 should be referenced in this early para in the Methods section (and moved to the Methods section). More importantly, Figure 2 should fully align with the Methods text and vice versa. This is currently not the case. E.g., the Methods do not seem to mention “modified delphi process” or “environmental scan” and the Figure does not mention some methods referenced in the text.
Lines 306-307. What are the “rapid qualitative analysis techniques” from “implementation science” that will be used? Some are later described, but not clear if all are (see above comment on Figure 2), and at each of the three steps. Please ensure all techniques used are clearly defined and described for each step.
Line 328. What are the “components” of BRACE mentioned here?
Line 337. Is “element” the same thing as “component”? Clarity in terms is needed.
Addition of the expected nature of outputs of the BRACE revision process would be useful, to the extent is can be provided. E.g., BRACE has been successful in part because of its simple stepwise approach. Does the study team anticipate retaining or changing the five steps? Will the process continue to be based on the same adaptive, iterative and stepwise approach that grounds BRACE? Etc. In addition, noting a general timetable in this study protocol would be helpful.
Reference list numbering is off in this draft as of ref 23.
Wishing the study team the best with this exciting and important work which will have great impact on the climate and health field, and on human health, in the coming years. Thank you for inviting me to review.
Author Response
REVIEWER 1
General Comments
- A clear one-sentence goal statement for this paper is currently lacking and is important to include, ideally in the introduction before the Methods section. It should also be present in the Abstract.
We have added a one sentence goal statement to the abstract and the end of the Introduction. Per the reviewers’ suggestions, we now focus this manuscript only on the revision of the BRACE framework and have removed the framework landscape analysis.
2a. Several long sections of the paper are without references and it is difficult to know on what basis the statements are being made, making these sections seem somewhat anecdotal rather than authoritative. Particularly lines:
- 138-168 (learning and evaluation of BRACE),
- lines 170-212 (landscape review of frameworks), and
- lines 229-281 (“enhancements”).
We thank the reviewer for identifying this. We have added references to lines 138-168 and 229-281 (lines note placement in the original draft). Per suggestions from both reviewers’, we have deleted the section on the landscape review of frameworks.
2b. The impression is of a viewpoint paper rather than a scholarly, data-driven evaluation and roadmap for review and revision. Detailed comments below provide some suggestions for remedying this. Two points are particularly notable:
- The landscape review of the climate adaptation framework literature reads almost like a separate independent paper, suggest authors consider this as an option. If so, that separate landscape review could be summarized briefly and reference in this paper.
Per the reviewers’ suggestions, we have removed the framework landscape component of this paper.
- The genesis of the four “enhancements” to BRACE is not provided. These four enhancements, which appear highly valid and worthy, form the grounding substantive guidance for the new BRACE framework. Yet the way in which these four were selected, and on what basis, is not evident from the paper. Why these four and not others?
These four enhancements were determined based upon feedback from current and former CDC Climate and Health Program recipients and literature review. This has been added to the methods, beginning on line 181 of the revised manuscript.
- This paper does not seem to consider several climate and health challenges that have recently grown substantially in importance for local US health departments. These include COVID-19 (lessons for local health departments engagement on climate, interactions with air quality, etc.), wildfire smoke exposure (in the US West and now East as well with Canadian wildfires), the major impacts of drought, particularly in the US West. Etc.
We have added these additional climate and health challenges to the Introduction, lines 48-51: “Recent events including the COVID-19 pandemic [10], wildfires and poor air quality [11,12], droughts [13], and extreme weather events [14,15] underscore the importance of recognizing the synergistic effects of climate change.”
- There is little attention to the distinction between urban and non-urban areas generally in the paper. Cities and states as well as tribes have been grantees under BRACE, and their capacities, resources, and experiences have differed. Given the large urban population affected by climate change (and the focus of many of the reviewed frameworks on cities), brief words on the urban experience, and specific framework-related issues for cities under the new BRACE would be helpful.
We agree that considerations between urban and non-urban areas are important for users of the revised BRACE framework. Our intention is to create a versatile framework with implementation guidance that is useful to many types of users, including those from urban and non-urban areas. We have added this and clarified that more contextual information about the experiences of urban and non-urban areas is a key consideration for the revised framework, the implementation guidance, and future research.
- The Methods section could be more clearly presented and organized. For example, Figure 2 belongs to this section, however, appears in the Introduction, and is currently not referenced or discussed in the paper, while this Figure and the Methods text are not mutually consistent.
We have revised the Methods section for clearer presentation and organization per the reviewer’s suggestion. Figure 2 has been revised to more clearly match the presentation in the Methods, moved to the appropriate place in the Methods (lines 257-260).
- What are the expected outputs and timeframe of this study protocol work? Will the iterative adaptive stepwise framework remain as an underpinning? Will the results be published, and in what way?
We have added a timeframe for this work in the Methods (lines 251-252). The work of the Expert Panel will determine the final structure of the revised framework, including the extent to which the new framework retains an iterative adaptive stepwise approach. Results of the final revised framework and associated implementation guidance tools will be published, presented at national conferences, made freely available on the CDC website and widely promoted once completed. We have added a brief section to the Discussion to describe this.
Specific Comments
Title
“Reimagining the role of PH departments… etc.” suggests the aspirational goal for PH departments to take on greater leadership for climate change policy and practice as a cornerstone of the new BRACE program. In this reviewer’s opinion, this vision is highly welcome and justified. However, the paper’s content does not sufficiently identify the reasons why PH departments are often sidelined from climate policy and practice currently; nor do the new pathways proposed for BRACE fully explain why public health departments might be in a different, better position with the new BRACE. Therefore, the paper does not seem to deliver yet on this title. Suggest a somewhat fuller diagnosis of why these departments currently are excluded from climate change efforts and how the proposed revised approach may change this, based on the large available literature (or modifying the title for a better fit with paper content).
Thank you for this suggestion. We appreciate the need for a full diagnosis of why health departments are currently excluded from climate change efforts, and we have added text on lines 82-96 with additional references pertaining to this issue (e.g. Mallen, et al. 2022; Ekstrom et al., 2014; Himmelstein et al., 2016; Maani & Galea, 2020 ). However, after clarifying our goal statement to reiterate the focus on protocol and methods, we feel that a lengthy description is beyond the scope of this paper. In the Discussion, we added that this issue should be further explored in future research.
Abstract. What is the goal of this paper?
A statement of the goal of the manuscript has been added to the abstract. Per the reviewers’ suggestion, the paper now focuses on the proposed methods for revising the BRACE framework.
Introduction
- Line 46-47.Would be useful to refer to air quality from wildfires which have substantially heightened climate and health awareness, particularly following COVID-19, and to the WHO estimates of annual deaths from air pollution. The climate and health effects in this section generally could be somewhat updated to reflect more recent learning and current/emerging concerns, including for example, widespread exposure to wildfire smoke.
We have added a brief description of these contemporary climate and health issues to the Introduction with appropriate references. See lines 48-51.
- Line 71-72. Many climate health effects are indeed felt locally. However, there are also others (like the above wildfire smoke example, but also for example drought) which may be felt regionally, and may require regional approaches. Given the growing challenges of air quality and water quality/quantity, incorporating into the new BRACE ways for local health departments to collaborate regionally on such topics may be of growing importance.
We have added appropriate references for these issues in the Discussion, as noted below:
“Although many of the health effects of climate change are felt locally, some effects such as smoke from wildfires are affecting large geographies (even across national borders), requiring regional adaptation/mitigation approaches. STLT health departments and their partners across the nation are recognizing the importance of collaborating regionally on these issues [66-68]. Future research should explore the utility of the revised BRACE in supporting regional collaborations and should also continue to examine the full suite of barriers and enablers pertaining to health departments’ support and promotion of climate change adaptation and mitigation efforts.”
- Line 76-73. It would be helpful to provide a clearer explanation of the problem stated that PH departments are “often not optimally aligned… etc.” and are “siloed, underfunded… etc.” The evidence from the literature for these statements is vast, but the citations for this literature do not seem to currently be presented in a helpful or specific way (with most added to the end of the para). Since this paper is intended to set out the approach to the new framework for BRACE, it should be authoritative and grounded in the literature.
We have provided a clearer explanation with additional references on lines 82-96, as follows:
“Most public health departments, particularly at the local level, face financial constraints [32]. Public health departments are often not optimally aligned to participate in larger community climate change planning, which may be led by a sustainability, environmental services and/or planning department, or an executive office [33]. Existing climate efforts within health departments are often siloed [34], underfunded [35,36], under resourced [37], and may lack capacity to adequately address inequities [38]. Public health departments frequently rely on grant funding that is designated for singular programmatic responses to specific infectious or chronic health conditions with limited latitude to address root causes or synergistic drivers of ill health [37]. Workforce capacity issues [39], and a general lack of environmental health literacy among the U.S. population further compounds these challenges [40]. To accelerate climate action in their jurisdictions, many health departments need supports to increase their internal capacity as well as collaborations across units, government departments and offices, and community organizations, especially those that represent people disproportionately affected by climate change [41-44].”
- Line 112-113. The difference between experience of states and cities in the BRACE program should be at least mentioned, and ideally somewhat expanded upon. This could carry through to the enhancements priorities and qualitative review for the new BRACE in subsequent sections of the paper -- i.e., will there be differences in approach and tools of the new BRACE framework for states vs cities (vs tribes). This is hinted at currently in the fourth "enhancement" priority area of "implementation in diverse settings," but not sufficiently developed as a theme.
We have added language to acknowledge the different experiences of states, cities, and tribal partners in the BRACE program on lines 168-169. We also added references pertaining to the experiences of BRACE grantees (Joseph et al., 2023; Mallen et al., 2022; Schramm 2020) and noted that creation of a versatile framework with appropriate implementation guidance for different user types, including states, cities and tribal partners as a goal of the revised framework (See section 2.2.4). Notably, the products resulting from this process (e.g., implementation guides) will aim to address some of these differences and may be tailored to reach a diverse set of end-users.
- Section 1.2. Identified gaps and other frameworks. This section would be more effective if it separated the two topics.
This section has been deleted from the manuscript per the reviewers’ suggestions.
- Lines 138-159.This section on “identified gaps” provides useful information on grantee experience. However, the whole section is unreferenced. Grant recipient feedback is generally mentioned as the source of these findings, including from phone calls and site visits. But no information is provided on the documented sources for these grantee feedback findings. Multiple documentation on BRACE grantee views have been made public over the years. Please add references to the documented sources for this feedback. Currently this section reads as too anecdotal for an authoritative piece, and does not provide enough clarity on where the “learning” is coming from. It is this reviewer’s understanding that CHP/BRACE has also been formally reviewed, and these formal reviews should also be mentioned within the “identified gaps” section. E.g., there was at least one expert workshop sponsored by CDC, to this reviewer’s knowledge, and likely more. The review findings should be mentioned, incorporated and referenced, even if briefly.
We have added citations published findings of formal reviews/evaluations of the BRACE Framework from the perspectives of CDC grant recipients.
- Lines 160-212. This section on “other frameworks” should be treated separately from “identified gaps” for clarity. This whole section is also unreferenced, and reads as somewhat anecdotal. Was the “landscape review” of climate adaptation frameworks conducted separately and produced as a report? If so, that report should be cited. Or, is this paper serving to report the findings for that landscape review? If this is the case, in essence then this paper has two goals, one a climate health framework study/review protocol and the other a landscape review of climate and health frameworks. At the very least this additional goal should be clarified in the manuscript goal statement, and the key findings reported in a way that is more concise and directly relevant to the main paper goal, i.e., the protocol for development of the new BRACE. However, another option the study team could consider would be to produce the climate and health frameworks landscape review as a separate review paper. Such a paper would allow expanded development of the findings of the landscape review which would be a good addition to the climate and health literature. That landscape review paper could then be briefly summarized and referenced in the present BRACE revision study protocol paper, with the outcome of helpfully sharpening the focus of the current BRACE revision protocol paper. The fact that the framework landscape review is an input to redesign of BRACE suggests it is well worth doing it as a comprehensive review and separate published manuscript.
We agree with the reviewer that the section on other frameworks was under-developed and did not add to the primary goal of this manuscript. We have deleted this from the manuscript.
- Line 184-186. The landscape review search string involved the keyword “adaptation” therefore it is not surprising adaptation was a core concept in the frameworks identified, and mitigation was not. The wording in these lines thus should be adjusted to make this clear. (But -- why not have broadened the search words to identify other frameworks not focused on adaptation only e.g., “health” and “mitigation”? Other frameworks dealing with mitigation might have emerged…)
Per the reviewer’s previous suggestion, we have deleted the landscape review from this paper.
- Importantly, and as noted for the Abstract, the paper currently lacks, and needs, a clear goal statement at the end of the introduction right before the Methods. “This paper aims to …”
A statement of the purpose of the manuscript has been added to the end of the Introduction (and the Abstract).
Methods
- The four “enhancements” to BRACE seem clear and relevant. But where did they come from?Why these and not others? A brief initial paragraph is needed to explain the genesis of these enhancement priorities, and the process of choosing these four from among many potential other enhancements. One gathers from the current text that this emerged from BRACE review recommendations, grantee feedback, review of the literature, review of other frameworks. But a short intro sentence/paragraph to describe the choice is important to make this clear.
The reviewer is correct in that the four enhancements were determined through a combination of CDC grant recipient feedback and emerging evidence/literature review. This has been added to lines 171-176.
- As a general comment, as noted above, additional clarity on state, city or tribal/territorial grantees and the differences in needs, challenges and conditions in each could be more adequately addressed in this paper.For example, in the “enhancements” one might expect to see some guiding principles for grants to cities, given many of the frameworks in the landscape review address urban areas.
As noted previously, we have added language to acknowledge the different experiences of states, cities, and tribal partners in the BRACE program on line 168-169. We also added references pertaining to the experiences of BRACE grantees (e.g., Joseph et al., 2023; Mallen et al., 2022; Schramm 2020). We have also clarified that the Expert Panel includes individuals that represent state, local and tribal perspectives (lines 268-277) and that the products resulting from this process (e.g., implementation guides) will aim to address some of these differences and may be tailored to reach a diverse set of end-users (section 2.2.4).
- Line 282. The methodological approach to revising the BRACE framework seems sounds and justified, and the specific qualitative methods to be appropriate. But this section is presented in a way that is somewhat difficult to follow currently. Specifically, Figure 2 (labeled as “methodology to develop, pilot test and disseminate the revised BRACE framework…”) is unanchored to the text, and appears in the Introduction rather than Methods section. Figure 2 should be referenced in this early para in the Methods section (and moved to the Methods section). More importantly, Figure 2 should fully align with the Methods text and vice versa. This is currently not the case. E.g., the Methods do not seem to mention “modified delphi process” or “environmental scan” and the Figure does not mention some methods referenced in the text.
We have revised Figure 2 and moved it to the Methods section. We have also made sure that the figure and text descriptions are aligned.
- Lines 306-307. What are the “rapid qualitative analysis techniques” from “implementation science” that will be used? Some are later described, but not clear if all are (see above comment on Figure 2), and at each of the three steps. Please ensure all techniques used are clearly defined and described for each step.
We have revised the Methods section (section 2.2) and Figure 2 to more clearly describe each of the methods that will be used to refine the framework and to create the implementation tools. We have more directly described the use of a Modified Delphi approach (including the 3 steps of the approach) with the Expert Panel.
- Line 328. What are the “components” of BRACE mentioned here?
The “components” are the anticipated parts/elements of the revised BRACE framework. The Expert Panel process will guide what is included in the new version of the framework. As it is unclear to what extent the original step-based approach will be retained (See response to point 7 below), we opted to refer to this as components.
- Line 337. Is “element” the same thing as “component”? Clarity in terms is needed.
Elements refer to the same thing as components. We have switched to use the term component consistently throughout the manuscript.
- Addition of the expected nature of outputs of the BRACE revision process would be useful, to the extent is can be provided. E.g., BRACE has been successful in part because of its simple stepwise approach. Does the study team anticipate retaining or changing the five steps? Will the process continue to be based on the same adaptive, iterative and stepwise approach that grounds BRACE? Etc. In addition, noting a general timetable in this study protocol would be helpful.
A key role of the Expert Panel is to provide feedback on the existing Framework and make suggestions for how to improve it. Thus, determination of what the specific “components” is currently being determined through the Methodology described in this manuscript. We have also added a general timetable to lines 251-252.
- Reference list numbering is off in this draft as of ref 23.
The references have been updated and corrected throughout the manuscript.
Reviewer 2 Report
REview of BRACE framework
Overall Comments and Questions:
Thank the authors for allowing me to review this timely and needed article in the literature.
This plan to revise the framework is definitely timely in the United States. It is important to also note that the work of the framework is very much a Northeast US lens, but still timely.
Question to the author about why not one local health department in their jurisdiction included on this manuscript and protocol plan? Was an invitation extended to the 20 local health departments(LHD’s) for inclusion in this work? Otherwise not truly community based or participatory.
I also would suggest that maybe partnering with EPA, NACCHO, or local and State (Department of Emergency Management and Homeland Security (DEMHS) and Departments of Environmental Protection (DEEP) would contribute a broader view of this work. It continues the siloed funding and work when all are not invited to at least review, give comment, seek areas of similarity.
Local Public Health Departments for many years in New England have had ESF’s already doing much of this work under the All-Hazards Approach for Emergency Preparedness at local, statee, and national levels. A suggestion to the authors is that a brief history of how the work has been done in the past or if published somewhere else would be good in the Introduction for the reading audience.
Introduction
Lines 74-78- Please check reference numbering through out the document, numbering seems to be off throughout the manuscript.
Lines 78-83- It is recommended that the authors change the statement and language here, they adequate site the problem but blame the victims. Local health departments for decades have suffered from unfunded and unresourced mandates that have decreased their capacity at many levels.
Recommend that table or figure be included to at least indicate the 20 health departments or their locations. It lends to affinity bias, since New England states might handle a forest fire a bit differently than California or the Midwest.
Figure 1. pg. 3. It is also a recommendation that evaluation be incorporated throughout the revised framework and not just at the end, this is not clear in the manuscript. Highlight the difference in the framework between process and program evaluation.
What is an estimated timeline for the revision work? Year? Two-years?
Was their any youth or volunteer engagement at the LHD level with entities such as Medical Reserve Corps (MRC’s) or similar entities? This would seem as though there is a great opportunity for youth engagement and volunteer engagement as well.
Lines 158-159- Suggest that authors do a better job for transition in writing about landscape review. Might also suggest that the authors consider taking it out and making it a separate manuscript.
Line 190- Supplemental materials have their own interpretation and implications as though they are an afterthought and not as important as the main body or work. Consider weaving health equity and suggestion for that work throughout the BRACE framework revision. Recommend that the authors consider limiting landscape review discussion and highlight and refine the work around incorporating health equity lens into BRACE framework revisions.
Line 191- 194 Suggest that authors move this statement possibly to limitations if landscape review is retained.
Lines 225-There is a statement that goes, “nothing about us without us”- this reviewer thinks that you have to make sure that your primary intended users have been included in the framework throughout otherwise it will end as so many other guidance and guidelines on a shelf at the LHD’s as unfunded mandate work.
Line 233- reference 33 should be reference 34, slight unevenness to the reference numbering seen throughout, please fix.
Line 351-363
As it stands now it seems as though the authors don’t truly understand how the work of everyday health departments gets done, because it is coming from an academic lens. Some of the language used in this (line 244-will be charged) creates more division and silos and not equity focused. Wonder if authors would consider an approach that assists LHD’s with infusion of BRACE principles and steps throughout their everyday work. A reimagining of the BRACE framework revisions to be inclusive snd connect with the everyday work and considerations of LHD’s. The revision might be an opportunity for a more comprehensive and integrative approach to the revision and integrationa dn implementation. Maybe have a local youth component tha could gamify the framework. I’ll stop there.
References:
48-53- don’t see where those references were used
Some references missing information-have access information be not general information, ex. #44, 47?
It was a pleasure to review the BRACE framework and suggestions for future revision.
Author Response
REVIEWER 2
Overall Comments and Questions
- Question to the author about why not one local health department in their jurisdiction included on this manuscript and protocol plan? Was an invitation extended to the 20 local health departments(LHD’s) for inclusion in this work? Otherwise not truly community based or participatory.
To date, the Climate and Health Program at CDC has funded 20 state and local health departments through its Climate-Ready States and Cities Initiative, along with 10 tribes through its Climate-Ready Tribes Initiative. Ten of these grant recipients are included on the Expert Panel that is informing the revision to the BRACE framework. Because the revised BRACE Framework is intended for use by CDC grant recipients as well as state, local, tribal and territorial health departments that don’t receive CDC funding and other potential users, we also included individuals on the Expert Panel who have not received CDC funding. A more detailed description of the Expert Panel members is presented in section 2.2.1.
- I also would suggest that maybe partnering with EPA, NACCHO, or local and State (Department of Emergency Management and Homeland Security (DEMHS) and Departments of Environmental Protection (DEEP) would contribute a broader view of this work. It continues the siloed funding and work when all are not invited to at least review, give comment, seek areas of similarity.
We recognize the importance of partnerships/collaborations for both the revised BRACE Framework and for health department involvement in climate-related work more broadly. The Expert Panel includes individuals who represent other federal agencies (n=2) and national organizations (n=8) addressing climate change to bring this perspective. That said, resource constraints prevent us from including representation from all such potential agencies/organizations. This is now noted as a limitation in the Discussion (lines 391-394).
- Local Public Health Departments for many years in New England have had ESF’s (Emergency Support Functions) already doing much of this work under the All-Hazards Approach for Emergency Preparedness at local, state, and national levels. A suggestion to the authors is that a brief history of how the work has been done in the past or if published somewhere else would be good in the Introduction for the reading audience.
Thank you for this suggestion. We acknowledge that Public Health Departments in the New England region have a history of ESF’s (Emergency Support Functions) performing much of this work under the All-Hazards Approach for Emergency Preparedness and have added a reference showcasing that approach (Wellenius et al. 2017). However, because the paper is intended for a national audience, we felt that it was beyond the scope to add a lengthier discussion focused only on New England, which may not be generalizable to other states.
Specific Comments
- Lines 74-7. Please check reference numbering throughout the document, numbering seems to be off throughout the manuscript.
Thank you for catching this. We have corrected and updated the references throughout the manuscript.
- Lines 78-83. It is recommended that the authors change the statement and language here, they adequate site the problem but blame the victims. Local health departments for decades have suffered from unfunded and unresourced mandates that have decreased their capacity at many levels.
As mentioned previously in the response to Reviewer #1, we added text to lines 82-96 to make it clear that we recognize the structural challenges and are not blaming the victims.
- Recommend that table or figure be included to at least indicate the 20 health departments or their locations. It lends to affinity bias, since New England states might handle a forest fire a bit differently than California or the Midwest.
We now cite the CDC Climate and Health Program website that shows their grant recipients (reference 46). The revised BRACE Framework is intended for use by future grant recipients as well as state, local, tribal and territorial health departments that do not have CDC funding. Because of this, we did not include a table of current and past CDC grant recipients.
- Figure 1. pg. 3. It is also a recommendation that evaluation be incorporated throughout the revised framework and not just at the end, this is not clear in the manuscript. Highlight the difference in the framework between process and program evaluation.
We agree that evaluation is important and should be incorporated throughout the revised Framework. Figure 1 shows the current BRACE Framework, which did visually represent evaluation at the end of the cycle. The Expert Panel process will inform the integration of evaluation into the revised Framework, paying attention to best practices that suggest evaluation is integrated throughout the process.
- What is an estimated timeline for the revision work?
The proposed timeframe for the revision work is two years. This has been added to lines 251-252.
- Was their any youth or volunteer engagement at the LHD level with entities such as Medical Reserve Corps (MRC’s) or similar entities? This would seem as though there is a great opportunity for youth engagement and volunteer engagement as well.
We agree that there is great opportunity for youth engagement and volunteer engagement in the user/implementation of the revised BRACE framework. We do not; however, have the resources to include youth in the Expert Panel. This is included as a limitation (See lines 391-393).
- Lines 158-159. Suggest that authors do a better job for transition in writing about landscape review. Might also suggest that the authors consider taking it out and making it a separate manuscript.
The landscape review has been removed from this manuscript.
- Line 190. Supplemental materials have their own interpretation and implications as though they are an afterthought and not as important as the main body or work. Consider weaving health equity and suggestion for that work throughout the BRACE framework revision. Recommend that the authors consider limiting landscape review discussion and highlight and refine the work around incorporating health equity lens into BRACE framework revisions.
We thank the reviewer for this comment, and we agree that the supplemental materials and the integration of equity are very important components of the revised BRACE framework. In fact, this is one of the major goals of the revision. We have clarified some of the language about the process, and we added a reference (St. George, 2023) to emphasize that an equity lens is part of our approach. We have centered equity in all phases of the project. For example, the Expert Panel was intentionally chosen to ensure that persons representing diverse communities were represented as we revised the BRACE framework. Per the reviewer’s suggestion, we have also deleted the landscape review from this manuscript.
- Line 191- 194. Suggest that authors move this statement possibly to limitations if landscape review is retained.
Per the suggestions of both reviewers’, the landscape review has been deleted from this manuscript.
- Lines 225. There is a statement that goes, “nothing about us without us”- this reviewer thinks that you have to make sure that your primary intended users have been included in the framework throughout otherwise it will end as so many other guidance and guidelines on a shelf at the LHD’s as unfunded mandate work.
The intention of using an Expert Panel led methodology is to develop a revised Framework and associated implementation guidance tools that are informed by the intended users and thus helpful to their future work. We have provided more information on the Expert Panel to lines 268-277.
- Line 233. reference 33 should be reference 34, slight unevenness to the reference numbering seen throughout, please fix.
We have corrected and updated the references throughout the manuscript.
- Line 351-363. As it stands now it seems as though the authors don’t truly understand how the work of everyday health departments gets done, because it is coming from an academic lens. Some of the language used in this (line 244-will be charged) creates more division and silos and not equity focused. Wonder if authors would consider an approach that assists LHD’s with infusion of BRACE principles and steps throughout their everyday work. A reimagining of the BRACE framework revisions to be inclusive and connect with the everyday work and considerations of LHD’s. The revision might be an opportunity for a more comprehensive and integrative approach to the revision and integration on implementation. Maybe have a local youth component that could gamify the framework. I’ll stop there.
Thank you for this suggestion. Because the primary focus of this paper is methodological, we feel that a more academic perspective is appropriate. However, we agree with the reviewer that for the process of revising BRACE as a whole, the lenses of practitioners from a variety of settings (including LHDs, tribal partners, and other collaborating organizations) are very important, We will include the voices of LHDs and other partners on the Expert Panel in all of the methods. We are including these voices as we iteratively build out the implementation guidance and supporting documents. However, resource and time constraints prevent us from including full representation from all such potential agencies/organizations, and we are unable to add a youth component at this time. These issues have now noted to the limitations in the Discussion (lines 393-394). We also state that these are important areas to consider for future research, and we added a reference (Wolff et al., 2017) that speaks to this.
- References.48-53- don’t see where those references were used, Some references missing information-have access information be not general information, ex. #44, 47?
We have corrected and updated the references throughout the manuscript.
Round 2
Reviewer 1 Report
I believe the study team has done a great job incorporating the peer review comments, and I have no further comments. Good luck to the team with this important revision to the BRACE program, which will have a major impact on climate health equity efforts in the US over the next years.
Reviewer 2 Report
The revisions have enhanced the paper.